# Effect of Polyphenylene Sulphide Particles and Films on the Properties of Polyphenylene Sulphide Composites

**DOI:** 10.3390/ma15217616

**Published:** 2022-10-29

**Authors:** Zeyu Sun, Li Sun, Chengyan Zhu, Wei Tian, Lingda Shao, Xuhuang Feng, Kunzhen Huang

**Affiliations:** 1Key Laboratory of Advanced Textile Materials and Preparation Technology of the Ministry of Education, College of Textiles Science and Engineering, Zhejiang Sci-Tech University (Xiasha Campus), Hangzhou 310018, China; 2Zhejiang Sci-Tech University Huzhou Research Institute Co., Ltd., Huzhou 313000, China; 3Hangzhou Pulay Information Technology Co., Ltd., Hangzhou 310016, China

**Keywords:** glass fibre, polyphenylene sulphide, composite laminate, dynamic mechanical analysis, flexural properties, impact properties

## Abstract

Glass fibre-reinforced polyphenylene sulphide composites were prepared by hot-pressing glass fibre fabrics and polyphenylene sulphide resins. The effects of different polyphenylene sulphide resin forms on the properties of the composites were investigated using scanning electron microscopy, dynamic mechanical analyser, pendulum impact tester and universal testing machine. The results showed that different polyphenylene sulphide resin forms had nearly no effect on the glass transition temperature of the composites, which are all located at about 100 °C. Compared with other polyphenylene sulphide composites, the bending strength of polyphenylene sulphide film composites was the highest, reaching 314.58 MPa, and the impact strength of polyphenylene sulphide particle composites was the highest, reaching 245.4 KJ/m^2^. The bending strength and impact strength were calculated using a standard fraction, and the highest standard fraction was obtained when the ratio of polyphenylene sulphide film to particle was 1:2. The impact strength and bending strength could be obtained. The impact strength reached 229.8 KJ/m^2^, and the bending strength reached 284.16 MPa.

## 1. Introduction

Composite materials are popular for their high specific stiffness and strength and are widely used in aerospace, transportation, sports equipment, wind power generation, medical equipment and other fields [1,2]. Among them, advanced composite materials represented by resin matrix composites have become the favourites of many applications, and their replacement of metal materials is an important development trend in the world manufacturing industry today. In a broad sense, resin-based composites can be divided into thermosetting resin-based composites and thermoplastic resin-based composites [3]. Compared with thermosetting resin-based composites, thermoplastic resin-based composites have many advantages: (1) thermoplastic resins only produce physical changes of softening by heat and hardening by cooling during the heating process without chemical reactions, so they can be recycled again and again, which meets the requirements of green and sustainable development in today’s society [4,5,6]; (2) thermoplastic resin-based composites have shorter moulding cycles, high efficiency, low cost and easy mechanical automation [7]; (3) thermoplastic resins have linear molecular chains with good toughness and impact resistance; and (4) thermoplastic resins demonstrate low moisture absorption, good resistance to heat and humidity and can continue to be used in high temperature and high humidity environments without significant degradation in performance [8,9]. Therefore, thermoplastic resin-based composites have received a lot of attention from scholars in recent years.

Polyphenylene sulphide (PPS), as a thermoplastic resin, has a molecular structure consisting of benzene rings interacting with sulphur atoms in a neat configuration, which makes it easy to form a crystalline structure with high thermal stability [10]. At the same time, its molecular structure causes PPS materials to have highly stable chemical bonding properties, and the benzene ring structure provides PPS with a large rigidity. In contrast, the thioether bond (-S-) provides a certain degree of flexibility, and the unique molecular structure of PPS materials has a number of properties superior to other thermoplastic resins. Its thermal properties are excellent, with short-term heat resistance up to 260 °C and long-term use at 200–240 °C [11]. The dielectric constant and dielectric loss angle tangent values of PPS are low and vary less over a wide range of frequencies and temperatures [12]. In addition, it has properties such as lower water absorption, better flame retardancy and better chemical resistance [13,14,15]. However, the high processing temperature and viscosity of the PPS resin matrix [16,17] have caused many difficulties in preparing high-performance fibre-reinforced thermoplastic composites.

Therefore, to expand the application of PPS composites, the key is to develop a suitable processing process. Samoryadov, A.V. et al. [18] studied the effect of the percentage of PPS and glass fibres in glass fibre-reinforced PPS composites on the physicomechanical, electrical and thermophysical properties, but only obtained the optimal percentage of glass fibres and PPS performance samples, not on the processing process of the in-depth composite study was conducted. Grouve, W.J.B., et al. [19] studied the composite curing model of glass fibre-reinforced PPS composites and used it as a guide for practice. Jamin, T. et al. [20] investigated the process of preparing carbon fibre-reinforced PPS composites, derived optimal process parameters for carbon fibre/polyphenylene sulphide stamping and moulding, and developed a nonlinear finite element model of the test specimens. However, their samples were made by the stamping and moulding method rather than the more common hot pressing method. Ma, Y. et al. [21] prepared self-reinforced PPS composites with different contents of PPS fibres by hot pressing method and investigated the effect of the ratio of PPS fibres to PPS resin in PPS composites on the tribological and thermal properties, but the mechanical properties, such as bending strength and impact strength of the composites, were not addressed.

There are few studies on the effect of PPS particles as well as film percentage on the properties of composites. Different PPS forms have different binding properties to glass fibres, which directly affects the mechanical properties of the composites, so it is of interest. In this paper, PPS particles and films were selected as the resin base of the composites, and the fibre-reinforced composite laminates were prepared using plain glass fibre fabrics soaked with silane coupling agent KH560 as the reinforcement base of the composites. The properties of glass fibre-reinforced PPS film/pellet composite laminates are systematically investigated, and their microstructures are observed.

## 2. Materials and Methods

### 2.1. Raw Materials

Polyphenylene sulphide pellets with a density of 1.35 g/cm^3^ were supplied by Zhejiang Xinghecheng Co., Ltd (Hangzhou, Zhejiang, China, for type 3514). The density of polyphenylene sulphide film is 7.43 g/cm^3^, supplied by Toray Group (Japan, for Toray Lina®Torelina® type). Glass fibre cloth with a gram weight per square meter of 400 g/m^2^ and a warp and weft density of 32 × 33 rods/10 cm was supplied by Jusco Group Co (Hangzhou, Zhejiang, China). Acetone was supplied by Siron Scientific (Hangzhou, Zhejiang, China, purity ≥ 99.8 %). Silane coupling agent model KH560 was supplied by Keisei Chemical Company (China, purity ≥ 98.0 %). Glacial acetic acid was provided by Xing Hua Chemical Company (China, purity ≥ 99.5 %). The thickness of polyimide release paper is 0.1 mm, provided by Hangzhou Mick Chemical Instrument Co (Hangzhou, Zhejiang, China).

### 2.2. Experimental Details

#### 2.2.1. Glass Fibre Plain Fabric Treatment

Untreated-GF: China JuShi Group glass fibre plain weave fabric. Marked as UGF.

Bare-GF: Configure acetone/anhydrous ethanol solution with a volume ratio of 3:7. UGF was cut to the correct size along the warp and weft direction and soaked in the solution at 30 °C for 48 h to remove the sizing agent and oil stains attached to the surface of the glass fibre fabric during the production process [22]. After removal, the fabric was repeatedly soaked and rinsed with ultrapure water more than five times to remove the residual acetone on the surface of the glass fibre and dried in a vacuum drying oven (Shanghai Jinghong Experimental Equipment Co.) at 80 °C for 12 h for use. That is called BGF.

Modified-GF: A silane coupling agent solution of silane coupling agent: ethanol: ultrapure water = 2:78:20 was prepared, and glacial acetic acid was added to adjust the PH to 5 [23,24,25,26]. The BGF was soaked in the silane coupling agent solution at 30 °C for 1 h. The silane coupling agent KH560 formed a single molecular layer or 2 or 3 molecular layers on the surface of the glass fibre fabric, and the -R-OH end group of the PPS resin would react with the epoxy group of the silane coupling agent KH560, thus combining the silane coupling agent with the glass fibre fabric [27]. The silane coupling agent tightly bonds together the PPS and glass fibre fabric. At the same time, due to the physical adsorption, the surface roughness of the initially smooth glass fibre fabric increases, the specific surface area increases, and the PPS is more easily bonded with the glass fibre fabric [27]. After being removed, the fabrics were washed with ultrapure water and dried in a vacuum oven at 80 °C for 12 h. The surface area of the fabrics was increased, and the PPS bonded more quickly to the glass fibre fabrics. Figure 1 shows a schematic diagram of the glass fibre modification preparation process.

#### 2.2.2. PPS Treatment

Figure 2 shows a top view of the PPS pellet and film. Cut the PPS film to a suitable size, wipe the surface with anhydrous ethanol for dust and stains, and place it in a vacuum drying oven at 80 °C for 12 h. Weigh the appropriate PPS pellets and put them in a vacuum drying oven at 80 °C for 12 h.

#### 2.2.3. Preparation of Glass Fibre-Reinforced PPS Composites

In this paper, glass fibre-reinforced PPS composites were prepared by the lamination method. MGF and PPS resin was laminated in the mould in turn, and the glass fibre fabric lay-up direction was [0°/90°]. The mould was placed in a semi-automatic plate vulcanizer (Wuxi Zhongkai Rubber Machinery Co., Ltd, model QLB-25 T type) (Wuxi, Jiangsu, China), set at 230 °C as the starting point, and the temperature was continuously increased at a rate of 3 °C/min for 30 min and heated to 320 °C for 20 min to prepare a 3 mm composite laminate. Figure 3 shows the structure of the composite laminate, which is named “P0F6”, “P2F4”, “P4F2” and “P6F0”, respectively, according to the variation of the percentage of PPS particles to PPS film, “P6F0”. The PPS films of “P2F4” and “P4F2” are set on the outer side to effectively improve the flatness of the surface of the composite laminate.

Figure 4 shows the lay-up diagram of the thermoforming. The top layer 2 and the bottom layer 4 are polyimide films, and the middle layer 3 is PPS resin and glass fibre fabric laminated sequentially, according to Figure 4.

### 2.3. Testing and Characterization

#### 2.3.1. Infrared Spectrum Test

PPS pellets and PPS film were soaked in acetone solution at 30 °C for 12 h to remove oil and other impurities from the surface of the PPS pellets or film. The films were repeatedly soaked and rinsed with ultrapure water more than five times to remove residual acetone and dried in a vacuum oven at 80 °C for 12 h. The molecular structure and chemical composition of the PPS pellets and films were observed using a Fourier infrared spectrometer (American Thermoelectric Corporation, USA, model Nicolet 5700) to analyse the differences between the films and pellets.

#### 2.3.2. Dynamic Mechanical Analysis

Referring to GB/T 40396-2021 “Experimental method for glass transition temperature of polymer matrix composites Dynamic mechanical analysis (DMA)”, the composites were cut into 1.4 cm wide and 3.5 cm long samples along the warp and weft direction of the composites, and dynamic mechanical tests were performed on the composites using a dynamic mechanical analyser (TA Instruments, DMA Q800) (American). The test was performed in a three-point bending mode, with air as the gas of choice, a test frequency of 1 Hz, an amplitude of 10 μm, a test temperature range of 20 °C to 200 °C and a heating rate of 5 °C/min [28,29].

#### 2.3.3. Bending Performance Test

Figure 5a shows the universal testing machine used to test the bending strength. Referring to GB/T 1449-2005 “Experimental method for bending properties of fibre-reinforced plastics” [30,31], ten 2.5-cm-wide and 6-cm-long samples of the composite were cut along the warp and weft direction of the composite, and the bending strength of the composite was tested using a universal testing machine (MTS Industrial Systems Co., Ltd. model Landmark 370.25) (American). The average bending strength was calculated from the data. The loading speed was 20 mm/min. The general equation of bending strength was as follows:(1)$$\sigma_{f}=\frac{3P\cdot l}{2b\cdot h^{2}}$$
where $\sigma_{f}$ is the bending strength, MPa; P is the breaking load, N; b is the width of the sample, mm; and h is the thickness of the sample, mm.

#### 2.3.4. Impact Performance Test

Figure 5b shows the pendulum impact tester used to test the impact strength. Referring to GB /T 1043.1-2008 “Determination of impact properties of plastic simply supported beams (Part I: non-instrumented impact test)”, ten 1-cm-wide and 8-cm-long samples of the composite were cut along the warp and weft direction of the composite, and the impact strength of the composite was tested using a pendulum impact tester (Steel Research Nack Testing Technology Co., Ltd. model NI series) (China), and the average impact strength was calculated from the data. The average impact strength was calculated based on the data. The initial precession angle was 150°, the impact torque was 268 N·m, and the instantaneous velocity of the impact sample was 5 m/s. The impact performance test specimens included notched and unnotched samples. The results of the tests performed on the two types of samples were not significantly better than each other, but for the impact performance analysis of GF/PPS composites, the a_cU_ values measured in the unnotched sample were the most conservative, so the unnotched sample was used, and the data were calculated in this paper [32,33]. The general equation for the impact strength of the unnotched sample is as follows:
(2)$$a_{cU}=\frac{E_{c}}{h\cdot b}\times 10^{3}$$
where $a_{cU}$ is the impact strength of the unnotched supported beam, KJ/m^2^; $E_{c}$ is the energy absorbed during the damage of the modified specimen, J; h is the thickness of the sample, mm; and b is the width of the model, mm.

#### 2.3.5. Standard Score Calculation

The standard fractions of impact strength and bending strength of the composite are calculated and summed to select the best formula. The general equation for the standard fraction calculation is as follows:(3)$$Z=\frac{\sigma_{f-\mu_{b}}}{\sigma_{b}}+\frac{a_{cU}-\mu_{i}}{\sigma_{i}}$$
where *Z* is the overall standard score, μ_b_ is the overall average bending strength of the composite, *μ_i_* is the overall average impact strength of the composite, *σ_b_* is the standard deviation of the bending strength and *σ_i_* is the standard deviation of the impact strength.

#### 2.3.6. Morphology Observation

A scanning electron microscope (Zeiss Group, Germany, model Sigma 300) (American) with an accelerating voltage of 3 kV and a working distance of 10 mm was used to observe the cross-sectional morphology of the composite and the bonding to the composite. For surface examination, a 500× magnification was used, and the samples were sputtered and gold-plated before observation.

## 3. Results and Discussion

### 3.1. Infrared Spectrum Analysis 

From Figure 6, the absorption peaks of the IR spectra were attributed as follows [34,35]: 1570, 1457 and 1381 cm^−1^ for the in-plane contraction vibration peak of C-C of a benzene ring; 1005 cm^−1^ for the in-plane deformation vibration peak of C=CH of the benzene ring; 1094 cm^−1^ for the in-plane expansion vibration peak of C-S; 1290 cm^−1^ for the in-plane deformation vibration peak of C-CH, and the out-of-plane deformation vibration peak of benzene ring C-H at 803 cm^−1^; the out-of-plane deformation vibration peak of benzene ring C-C at 740 cm^−1^; the S=O bond stretching vibration peak of sulfoxide at 1075 cm^−1^; and the -SO_2_^−^ bond asymmetric stretching vibration peak and symmetric stretching vibration peak of sulfone at 1259 and 1182 cm^−1^, respectively. The PPS resin was verified to be chemically similar to the film according to the absorption peaks of IR spectra. 

The sulfoxide characteristic peak at 1075 cm^−1^ indicates that there is a certain degree of thermal oxidation of the film and the pellets, and the production of PPS pellets and films is carried out in a high-temperature environment, during which some of the polyphenylene sulphides undergo high-temperature oxidation, resulting in the formation of sulfone bonds, which further increases the brittleness of the polyphenylene sulphide [36]. Furthermore, the sulfoxide-based stretching vibrational peak at 1075 cm^−1^ was higher than those of PPS particles. This is because PPS pellets are prepared by extruding the PPS resin through a twin-screw extruder followed by cooling and pelletizing, while films are produced by extruding the resin through a twin-screw extruder and initially shaped through a T-shaped groove, then stretched longitudinally and transversely and shaped by high-temperature heat treatment [37,38]. Compared to pellets, films have a longer processing time at high temperatures, and the S=O bond and -SO_2_- characteristic peaks appear and increase as the oxidation time at high temperatures increases.

### 3.2. Effect of Polyphenylene Sulfide Morphology on Dynamic Mechanical Properties of Composites

Figure 7 shows the dynamic thermodynamic test plots for composite laminates with different PPS resin percentages. Figure 7a shows that the energy storage modulus of PPS film composites is higher than that of PPS particle composites throughout the tested temperature range, especially before the glass transition temperature. The energy storage modulus of all samples decreases with the increase in temperature because the mobility of polyphenylene sulphide molecular chains increases with the increase in temperature [39]. The composite’s energy storage modulus decreases with the percentage of polyphenylene sulphide particles, and the energy storage modulus of all samples decreases with the increase in temperature. The energy storage modulus is essentially Young’s modulus, which is an indicator of the rebound of material after deformation and indicates the ability of the material to store elastic deformation energy. Generally, higher crystallinity corresponds to a higher modulus [40]. Due to the film heat treatment process, the film molecular chain arrangement tends to tighten [41], and the crystallinity of PPS enhances.

Figure 7b shows the loss modulus curves of the composite laminates with different percentages of PPS particles and films. It can be seen from the figure that the loss modulus of the composite decreases as the percentage of PPS particles in the composite laminate rises. This is because the preparation process of PPS film has an additional heat treatment compared to PPS pellets [42]; this process increases the crystallinity of the PPS film but leads to low fluidity of the film itself, insufficient bonding properties between the film and the glass fibre fabric and the existence of voids [43]. The losses modulus of P2F4 and P4F2 are similar because the PPS pellets in the molten state fill the bonding voids between the PPS film and the bonding void between the glass fibre fabric, effectively compensating for the poor bonding between the film and the glass fibre fabric.

The first tangent line is made at the flat before the transition of the energy storage modulus. The inflexion point is selected at the decreasing section of the energy storage modulus through the first-order differential curve. The second tangent line is made at the inflexion point. The temperature corresponding to the intersection of the first and second tangent lines is several bits of glass transition temperature (Tg). Table 1 shows the composite’s inflexion point and glass transition temperature. As seen from the table, with the transformation of the polyphenylene sulphide form, the Tg of the composites did not show significant changes, all located around 100 °C.

### 3.3. Effect of Polyphenylene Sulfide form on Mechanical Properties of Composites

Figure 8 shows the effect of different forms of PPS on the impact and bending properties of the composites, and it can be seen that the impact strength of the composites gradually increases as the percentage of PPS particles rises. The impact strength of P6F0 reached 245.4 KJ/m^2^, which is 72.4 % higher than that of P0F6. The glass transition temperature (Tg) of PPS resins is about 90 °C [44]. 

PPS molecular chains increase activity and kinetic energy at temperatures above Tg and within a certain time frame and undergo migration and rearrangement, which promotes the movement of grain boundaries and grain growth, resulting in an effective increase in the crystalline integrity of the material. The degree of crystalline integrity of the material is effectively improved, leading to an increase in the crystallinity of PPS and further refinement of the crystalline morphology [45]. During the heat treatment process, the crystallinity of PPS rises, resulting in a tighter arrangement of its molecular chains and a decrease in porosity, leading to a composite material with no room for molecular chain segments to move upon impact and a decrease in impact strength [46]. Therefore, as the percentage of thin films in the composite PPS resin rises, the impact strength of the composite material decreases.

The bending strength of the composites was P0F6 > P4F2 > P2F4 > P6F0, and the bending strength of the composites did not end up decreasing with the increase in the percentage of particles. The bending strength of P0F6 was higher than that of P6F0 because the heat treatment process of the films increased the crystallinity of PPS and made the molecular chains of the films more closely aligned, which resulted in the bending strength of the PPS films, and their prepared composites increased by 13.13% [47]. The bending strength of PPS films and their prepared composites was enhanced. The bending strength of P4F2 reached 284.16 MPa, which is 13.13% higher compared to 251.17 MPa of P2F4. Figure 9 shows the SEM cross-sectional view of the impact fracture samples of glass fibre-reinforced PPS composites. Figure 9a shows the impact fracture interface between the polyphenylene sulphide particles and the glass fibre fabric; Figure 9b shows the impact fracture interface between the polyphenylene sulphide film and the glass fibre fabric. The fractured glass fibres and PPS particle resin bonding interface showed almost no holes; under the impact effect, the fractured glass fibres are wrapped or semi-wrapped and did not expose the fibre roots; the particle resin impact fracture surface was uneven with a large specific surface area, so that the resin and glass fibre contact surface was large, with a strong bond between the two. The fractured glass fibre and PPS particle resin bonding interface demonstrated the impact of a large number of glass fibres exposed, indicating that the film resin and glass fibre bonding force is poor; glass fibres between the existence of gaps not filled by PPS film resin indicated that in the hot pressing process, PPS film resin mobility is poor and not completely wrapped around the glass fibre; PPS film resin on the fracture surface exposed by the impact is flatter, in contrast to the PPS particle resin. The above phenomenon explains the high impact strength of the composite laminate with a high percentage of PPS particles. Figure 9c shows the impact fracture interface between PPS pellet resin and PPS film resin. The PPS particles and film are well bonded in the figure, and there is no significant delamination between the two at the bonding interface between the glass fibre fabrics. This indicates that the PPS particles and films can be well bonded during the hot pressing of the composite, and the bonding of the PPS particles and films has a negligible effect on the interlayer structure of the composite.

The bending strengths of the composites were P0F6 > P4F2 > P2F4 > P6F0; the bending strengths of the composites did not decrease ultimately with the increase in the percentage of particles. The bending strength of P0F6 was higher than that of P6F0 because the heat treatment process of the films enhanced the crystallinity of PPS and tightened the molecular chain arrangement of the films, which enhanced the bending strength of the PPS films and their prepared composites by 13.13 % [47]. The bending strength of PPS films and their prepared composites was enhanced. The bending strength of P4F2 reached 284.16 MPa, which is 13.13 % higher than the 251.17 MPa of P2F4. Figure 9 shows the SEM cross-sectional images of the impact fracture samples of glass fibre fabric-reinforced PPS composites; in Figure 9a, the fractured glass fibres in contact with polyphenylene sulphide have less fracture gap, and the fractured glass fibres are all wrapped or semi-wrapped. In Figure 9b, there is a large gap between the fracture surface of polyphenylene sulphide film and glass fibres, and part of the fractured glass fibres are exposed to the surface. This shows that the binding performance of polyphenylene sulphide particles to glass fibres is better than that of polyphenylene sulphide films. The reason for this phenomenon is that the heat treatment of the film gives high crystallinity to the PPS resin while increasing the viscosity of the resin matrix and reducing its fluidity. In the hot pressing process of the composite, the film with high viscosity has poor mobility, and its infiltration to the glass fibre tows is insufficient, resulting in a decrease in the bending performance of the composite. In Figure 9c, the PPS particles and the PPS film are well bonded, and there is no apparent delamination between the two at the bonding interface between the glass fibre fabrics. It shows that the PPS particles and the film can be well integrated during the hot pressing process of the composites, and the combination of PPS particles and the film has a negligible effect on the interlayer structure of the composites.

Table 2 shows the standard fractions of bending and impact strength of P0F6–P6F0. P6F0 has the highest impact strength but the lowest bending strength, while P0F6 is the opposite. In this paper, the bending and impact strengths of the composites are converted to common fractions to make them comparable and addable and to reflect the differences in mechanical properties of different composites accurately. Table 2 shows that when the ratio of film to particle reaches 1:2, composites with better impact strength and bending strength can be obtained.

## 4. Conclusions

PPS resins are chemically similar to films, and the PPS resin’s form affects the PPS composites’ mechanical properties. The PPS pellets and the films are well integrated, and the effect of the different PPS forms on the composites’ interlayer structure is negligible. The storage modulus, as well as the loss modulus of the composites, increase with the percentage of PPS particles. The effect of the different forms of PPS on the Tg of the composites is almost absent and is located around 100 °C. The impact properties of the composites increase with the percentage of PPS particles.

In contrast, the flexural properties of the composites are affected by both the properties of PPS itself and the bonding properties of PPS and glass fibres. As the proportion of film increases, the high crystallinity of the film plays a role in improving the bending performance, and the lack of film and glass fibre bonding performance plays a role in reducing the bending performance under the joint action of the two factors when the ratio of film-to-particle reaches 1:2, the composite material with better impact strength and bending strength can be obtained. Its impact and bending strength reach 229.8 KJ/m^2^ and 284.16 MPa, respectively.

## Figures and Tables

**Figure 1 materials-15-07616-f001:**
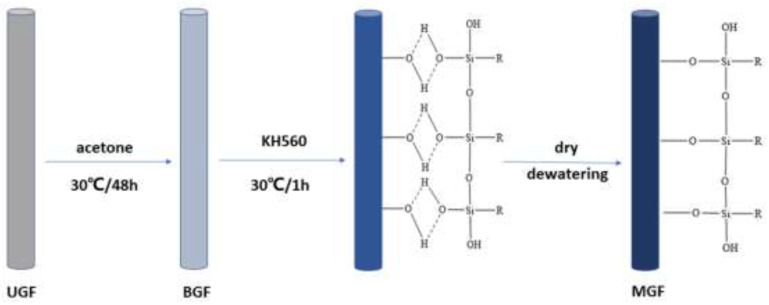
Glass fibre modification.

**Figure 2 materials-15-07616-f002:**
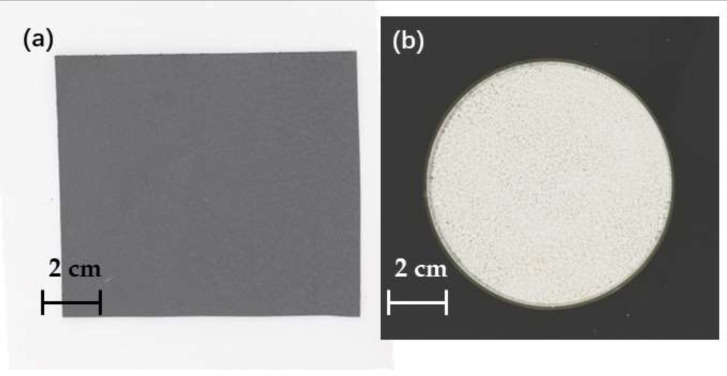
(**a**) Digieye top view shot of PPS film; (**b**) Digieye top view shot of PPS particles.

**Figure 3 materials-15-07616-f003:**
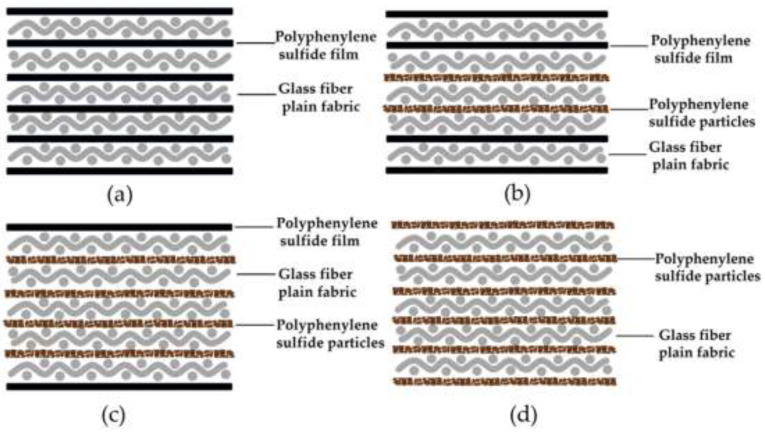
Composite laminate structure diagram. (**a**) P0F6, pure PPS film; (**b**) P2F4, the ratio of film to particle is 2:1; (**c**) P4F2, the ratio of film to particle is 1:2; (**d**) P6F0, pure PPS particles.

**Figure 4 materials-15-07616-f004:**
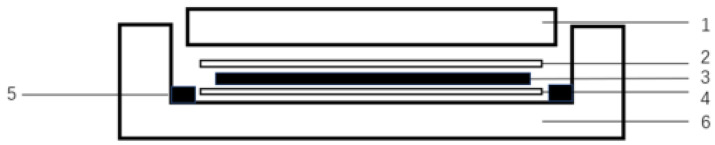
Hot pressing diagram. 1, upper cover plate; 2, polyimide film; 3, prefabricated parts; 4, polyimide film; 5, filler piece; 6, lower mould.

**Figure 5 materials-15-07616-f005:**
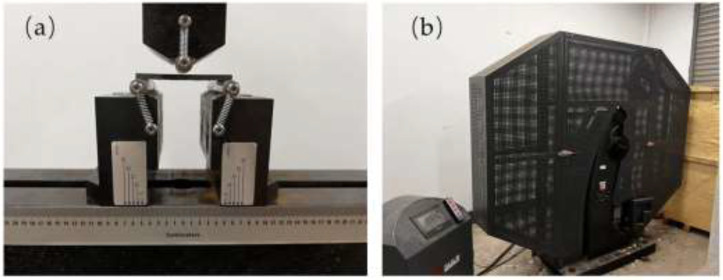
(**a**) Testing bending performance with the universal testing machine; (**b**) pendulum impact tester testing impact performance.

**Figure 6 materials-15-07616-f006:**
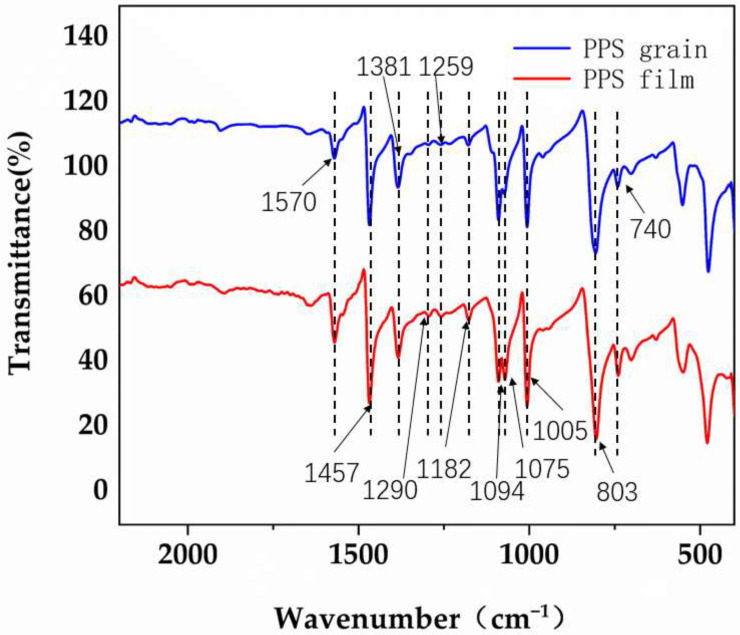
IR spectra of PPS in different forms.

**Figure 7 materials-15-07616-f007:**
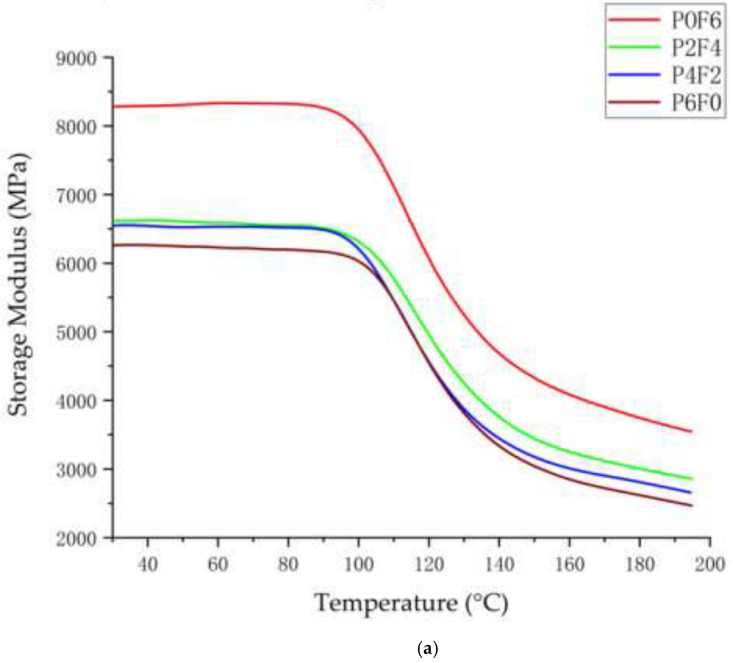
Effect of different polyphenylene sulphide forms on the dynamic mechanical properties of composites: (**a**) storage modulus; (**b**) loss modulus.

**Figure 8 materials-15-07616-f008:**
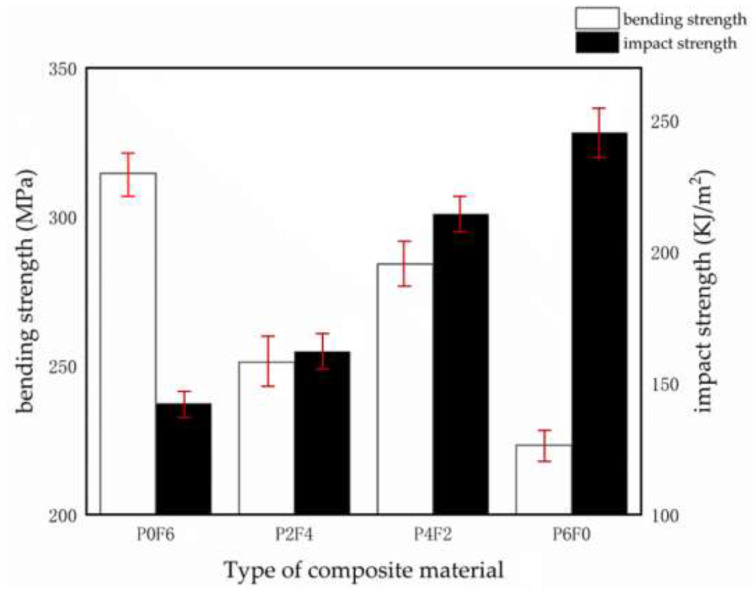
Effect of PPS form on mechanical properties of composites.

**Figure 9 materials-15-07616-f009:**
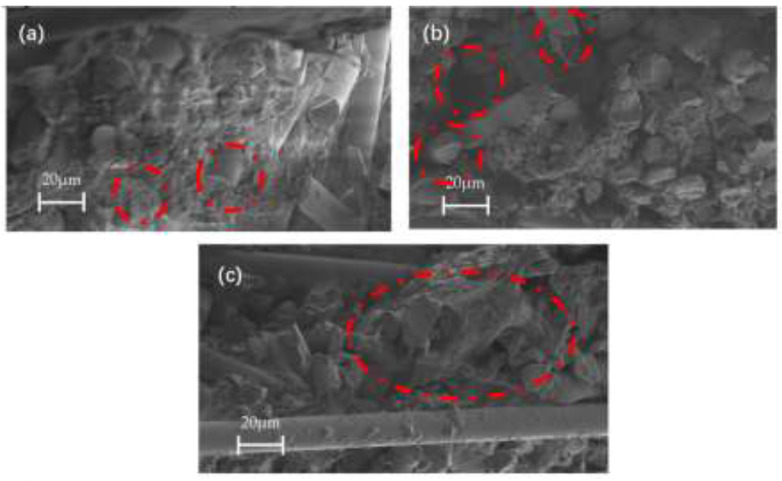
SEM cross-sectional image of impact fracture pattern of Glass Fibre Fabric Reinforced PPS Composite (**a**) PPS (particle) and glass fibre tow; (**b**) PPS (film) and glass fibre tow; (**c**) combination of film and particle.

**Table 1 materials-15-07616-t001:** Table of the glass transition temperature of composite materials.

	P0F6	P2F4	P4F2	P6F0
Inflexion point temperature (°C)	115 ± 1	116 ± 1	112 ± 2	116 ± 1
Tg (°C)	102 ± 2	100 ± 1	100 ± 1	102 ± 1

**Table 2 materials-15-07616-t002:** Composite material standard score.

	Bending Standard Score	Impact Standard Score	Total Standard Score
P6F0	−1.1354	1.1486	0.0132
P4F2	0.4001	0.4938	0.8939
P2F4	−0.4324	−0.6132	−1.0456
P0F6	1.1677	−1.0295	0.1382

## Data Availability

Not applicable.

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
