# Peer review of "Effect of Polyphenylene Sulphide Particles and Films on the Properties of Polyphenylene Sulphide Composites"

_materials, 2022, doi:10.3390/ma15217616_

Round 1

Reviewer 1 Report

This article discusses the mechanical performance of compressed sheets of PPS and glass fibres. There are several shortcomings that need to be addressed before the work is ready for publication, although I would judge the impact low.

Overall, by reading the paper there are two key issues:

1) Comparison of the present results with others in the literature is missing

2) The motivation is kind of not well-supported. Isn't it quite expected that films would have higher crystallinity and thus increased mechanical performance?

The expression "PPS form" is not fully in the way it is used. In this work, it refers to either PPS film, pellets, of combination of the two in the composite film. I feel that this should be clarified better (also in the abstract). Furthermore, the word "plate" is also used in the text interchangeably, which is somewhat confusing.

Page 1: Please rephrase "Therefore, scholars have received a lot of

attention from thermoplastic resin-based composites in recent years." to

"Therefore, thermoplastic resin-based composites have received a lot of

attention from scholars in recent years."

Please add scale bars to figure 2.

Also on figure 2: What is the information carried by the figure? Especially on the film, is the surface structure that is of interest, or something else?

The image quality needs to be improved in the following figures: 1, 3, 5, 6, 7, 8.

Figure 5: IR spectra of only two forms are shown; what about the other two?

Figure 6: Unfortunatelly it is not possible to tell which curve corresponds to which Plate. Also in the text (section 3.2 of the previous page), the modulus of Plate4 is mentioned twice, but of Plate1 not at all?

Please homogeneise accross the manuscript by inserting a space between the numerical value and unit symbol, wherever this is missing.

The references are not geographically well-spread.

Author Response

Dear Reviewer,

On behalf of my coauthor, I would like to thank you for giving me the opportunity to revise and resubmit my paper entitled "Effects of PPS particles and films on the properties of PPS". Your guidance is very helpful to our paper revision. We have carefully considered and responded to every proposal. The reviewers' comments and suggestions and our response to the reviewers' comments can be found in the pdf file.

Reviewer 2 Report

I have read the manuscript provided by the authors and I have to say that it needs a lot of improvement. There are many questions relating to scientific work and how results are presented. Moreover, the quality of almost all images is very bad. These images must be improved in terms of quality.

Finally, authors should provide specific references to support their claims in many cases.

Please see pdf attached.

Author Response

Dear Reviewer,

On behalf of my coauthor, I would like to thank you for giving me the opportunity to revise and resubmit my paper entitled “Effects of PPS particles and films on the properties of PPS”. Your guidance is very helpful to our paper revision. We have carefully considered and responded to every proposal.  The reviewers' comments and suggestions and our response to the reviewers' comments can be found in the pdf file.

Reviewer 3 Report

The present manuscript entitled “Effect of polyphenylene sulfide form on properties of Polyphenylene Sulfide Composites” authored by Zeyu Sun et al. describes the Glass fiber-reinforced polyphenylene sulfide composites were prepared by hot pressing glass fiber fabrics and polyphenylene sulfide resins. Furthermore, different polyphenylene sulfide resin forms on the properties of the composites were investigated using several characterization techniques. The bending strength, as well as the impact strength, were calculated using a standard fraction. The results exhibited that impact strength reached 229.8kJ/m2, and the bending strength reached 284.16MPa. The text does not contain major language mistakes. The objective and justification of the work are clear, and the experimental work is significant. The study is very accurate and adequate, and thus, I recommend it for publication. However, certain Minor issues are detailed below which need to be addressed before its final acceptance in the Materials.

Comment 1:  There are some typographical errors in the manuscript text, so the authors need to correct them in the revised manuscript.

Comment 2: The introduction is well written, and appropriate information is provided. However, include some more recent years’ literature in the introduction section to strengthen their work.

Comment 3: Figures 1, 5, 6, and 7,  quality is very poor, so provide the High-resolution figures.

Comment 4: The SEM results explanation should be discussed wider and compared with the other studies.

Comment 5: Check the reference style and maintain the journal names as abbreviations according to the MDPI format.

Author Response

(The authors gave the same response as above.)

Round 2

Reviewer 1 Report

The authors addressed the highlighted topics adequately.
Some figure are probably still of need for better resolution.

Author Response

Dear Reviewer,

On behalf of my coauthor, I would like to thank you for giving me the opportunity to revise and resubmit my paper entitled " Effect of polyphenylene sulphide particles and films on the properties of polyphenylene sulphide composites". Your guidance is very helpful to our paper revision. We have carefully considered and responded to every proposal. The comments and suggestions of the reviewers and our response to the reviewers' comments are as follows:

Comment:

Modifications are sought to respond to the comments listed below.

The authors addressed the highlighted topics adequately.

Authors response: Thank you for your comments, without your help in pointing out the shortcomings of the paper we would not have been able to improve it.

Some figure are probably still of need for better resolution.

Authors response: I replaced the images with the original high-resolution images. I turned off the image compression on the word document to do my best to ensure that the images were as straightforward as possible. At the same time, I communicated with the paper's editor to have it checked and to find a solution after uploading it.

We once again thank the anonymous reviewers for devoting their valuable time to give such appreciable insights.

Regards,

Zeyu Sun

Reviewer 2 Report

I have read the revised manuscript provided by the authors and I have to say that has improved a lot compared to the first version. However, I think they should revise some points before accepting the final form.

1) I think that the references are not in order. Some numbers appear before others.

2) Figure 2, still does not appear inside the text.

3) I have seen that the authors have improved the quality of the images, but it would be better if they can provide better images for figures 7 and 8

Author Response

Dear Reviewer,
On behalf of my coauthor, I would like to thank you for giving me the opportunity to revise and resubmit my paper entitled " Effect of polyphenylene sulphide particles and films on the properties of polyphenylene sulphide composites". Your guidance is very helpful to our paper revision. We have carefully considered and responded to every proposal. The comments and suggestions of the reviewers and our response to the reviewers' comments are as follows:
Comment:
Modifications are sought to respond to the comments listed below.
1)I think that the references are not in order. Some numbers appear before others.
Authors response: Thank you for pointing this out, we have checked and corrected the order of citations in the literature.

2) Figure 2, still does not appear inside the text.
Authors response: Thanks to your correction, we have described Figure 2 in 2.2.2.

3) I have seen that the authors have improved the quality of the images, but it would be better if they can provide better images for figures 7 and 8
Authors response: We have increased the DPI pixels from 600 to 1200 in Figure 7 as well as Figure 8, which we hope will make a difference. We also spoke to the thesis editor to prevent the images from being compressed and losing clarity during the upload process.

We once again thank the anonymous reviewers for devoting their valuable time to give such appreciable insights.
Regards,
Zeyu Sun
